# Public Profile Matters: A Scalable Integrated Approach to Recommend Citations in the Wild

## Abstract

Proper citation of relevant literature is essential for contextualising and validating scientific contributions. While current citation recommendation systems leverage local and global textual information, they often overlook the nuances of the human citation behaviour. Recent methods that incorporate such patterns improve performance but incur high computational costs and introduce systematic biases into downstream rerankers. To address this, we propose *Profiler*, a lightweight, non-learnable module that captures human citation patterns efficiently and without bias, significantly enhancing candidate retrieval. Furthermore, we identify a critical limitation in current evaluation protocol: the systems are assessed in a transductive setting, which fails to reflect real-world scenarios. We introduce a rigorous *Inductive* evaluation setting that enforces strict temporal constraints, simulating the recommendation of citations for newly authored papers in the wild. Finally, we present *DAVINCI*, a novel reranking model that integrates profiler-derived confidence priors with semantic information via an adaptive vector-gating mechanism. Our system achieves new state-of-the-art results across multiple benchmark datasets, demonstrating superior efficiency and generalisability. The code and the trained models will be made available upon acceptance.

## 1 Introduction

The rapid expansion of scientific research has led to an exponential surge in published literature (Drozdz & Ladomery, 2024; Rousseau et al., 2023). This information deluge presents a significant bottleneck for researchers attempting to identify and integrate relevant prior work (Datta et al., 2024; Bhagavatula et al., 2018). Consequently, there is a critical need for automated systems that can efficiently streamline the citation process (Goyal et al., 2024; Gu et al., 2022). Citation recommendation methodologies are generally categorised into two paradigms: "global" (Ni et al., 2024; Ali et al., 2021; Xie et al., 2021) and "local" (Jeong et al., 2020; Dai et al., 2019; Ebesu & Fang, 2017; Huang et al., 2015; Livne et al., 2014; He et al., 2010). While global recommendation suggests papers based on the overall theme of a document, local citation recommendation (LCR) operates at a fine-grained level, and is the focus of this research work. LCR targets specific "citation contexts" or excerpts, aiming to suggest references that align semantically and conceptually with the immediate narrative of a passage.

State-of-the-art (SOTA) LCR systems typically leverage metadata like titles and abstracts alongside citation contexts. For instance, SymTax (Goyal et al., 2024) utilises a three-stage architecture involving a prefetcher, an "enricher" to capture symbiotic neighbourhood relationships, and a reranker. However, this approach faces three major challenges. First, the enricher mimics human citation behaviour, i.e., specifically the tendency to cite from a narrow pool of seminal works, which while effective, introduces and perpetuates inherent "confirmation bias" in the citation ecosystem. Second, the three-stage candidate retrieval process imposes significant computational overhead. Third, its reliance on paper-specific taxonomy limits generalisability, as such metadata is often unavailable in benchmark datasets.

More recently, Çelik & Tekir (2025) proposed CiteBART to generate parenthetical author-year strings directly for an input citation context. We identify two critical flaws in this setup: (i) the generative nature leads to hallucinations of non-existent citations, and (ii) the framework is semantically decoupled from the

research content. By focusing on "author-year" strings, the model treats research as a function of primary authors' names (e.g., "Celik" or "Goyal") rather than the substantive scientific content, which is fundamentally independent of such identifiers. Moreover, we shed light on the current training and evaluation practice of LCR systems operating in a setting that deviates from real-world scenarios. To address these limitations, we make following contributions:

- We introduce the **Profiler**, a lightweight, non-learnable module for candidate retrieval. It is remarkably efficient and free from confirmational bias, yet it outperforms the sequential combination of prefetcher and enricher.

- We demonstrate the importance of a paper's "public profile", i.e., how the research ecosystem perceives a paper, as a remarkably vital signal for recommendation.

- We develop the **DAVINCI reranker**, which discriminatively integrates confidence priors with textual semantics via an adaptive vector-gating mechanism. Unlike previous SOTA, it is architecturally generalisable across diverse datasets without requiring special metadata like taxonomies.

- We establish a new state-of-the-art, demonstrating that DAVINCI surpasses both specialised LCR systems and massive-scale open-source rerankers adapted for this task.

- Finally, we introduce and benchmark LCR in an **Inductive setting**, providing a more realistic evaluation framework for citations "in the wild."

## 2 Related Work

Early investigations, such as that by He et al. (2010); Livne et al. (2014); Huang et al. (2015); Ebesu & Fang (2017); Dai et al. (2019), formally introduced local citation recommendation, utilising approachs ranging from TF-IDF based vector similarity to bidirectional LSTMs for modelling contextual information. In an effort to integrate both contextual signals and graph-based signals, Jeong et al. (2020) proposed the BERT-GCN model. This model leverages BERT (Kenton & Toutanova, 2019) to generate contextualised embeddings for citation contexts, capturing semantic nuances. Simultaneously, it employs a Graph Convolutional Network (GCN) (Kipf & Welling, 2017) extracting structural information from citation network, to determine the relevance between context and potential citations. However, as noted by Gu et al. (2022), the computational intensity inherent in GCNs posed a significant practical challenge. Consequently, the BERT-GCN model's evaluation was constrained to small datasets with only a few thousand citation contexts. This limitation emphasises a critical scalability bottleneck for GNN-based recommendation models when applied to large-scale datasets, highlighting the need for more computationally efficient techniques.

Medić & Šnajder (2020) explored the integration of global document information to enhance citation recommendation. However, as reported in Gu et al. (2022) and Goyal et al. (2024), it creates an artificial setup which in reality does not exist. Ostendorff et al. (2022) suggested a graph-centric approach (SciNCL), utilising neighbourhood contrastive learning across the complete citation graph to generate informative citation embeddings. These embeddings facilitate efficient retrieval of top recommendations using k-nearest neighbourhood indexing. Recently, Gu et al. (2022) introduced an efficient two-stage recommendation architecture (HAtten) which strategically separates the recommendation process into rapid prefetching stage and a more refined reranking stage, optimising for both speed and accuracy. Building upon HAtten, Goyal et al. (2024) proposed a three-stage recommendation architecture (SymTax) composed of prefetcher, enricher and reranker, establishing state-of-the-art in local citation recommendation. Very recently, Çelik & Tekir (2025) performed continual pre-training of BART-base to generate correct parenthetical author-year citation for a given context. Crucially, this generative approach relies heavily on author-year surface forms rather than the underlying research contributions. This creates a semantic bottleneck where the model prioritises bibliographic identifiers over the actual scientific content, which is inherently independent of the authors' identities. This is concerning, given that LLM policy of premier conferences like NeurIPS, ICLR etc. considers even one hallucinated citation to be a ground for a paper's rejection or revocation. Moreover, a very recent analysis of $4,841$ papers accepted in NeurIPS 2025 by GPTZero, uncovered 100s of hallucinated citations missed by the

3+ reviewers who evaluated each paper (Shmatko et al., 2026). Since NeurIPS 2025 had an acceptance rate for main track papers of 24.52%, each of these papers beat out $15,000$ other papers despite containing one or more hallucinations. With enough insights and motivation from the above discussion on state-of-the-art, we now discuss our proposed work in detail.

## 3 Proposed Work

### 3.1 Problem Formulation

We formulate the task of local citation recommendation as a two-stage retrieval and reranking problem, designed to handle the immense scale of modern scholarly corpora. Given a query instance $q = (S_q, M_q)$ — comprising a snippet of citation context $S_q$ and the source document's meta information $M_q$ characterised by its title $T_q$ and abstract $A_q$ — and a large corpus of scientific documents $C = \{D_i\}$, the process is as follows. First, in the **retrieval stage**, our novel **Profiler** module efficiently retrieves an initial candidate set $C_q \subset C$, where $|C_q| \ll |C|$. For each candidate document $c_i \in C_q$, Profiler also yields a confidence score, $s_i$, which serves as an initial estimate of its relevance. Second, in the **reranking stage**, our proposed **DAVINCI** module ingests this candidate set and their associated confidence scores. It computes a final, fine-grained relevance score, $f_{\mathrm{DAVINCI}}(q, c_i, s_i)$, by fusing a deep semantic analysis of the content with the discriminative priors obtained by refining the confidence signal from the Profiler. The final output is a ranked list $L_q$ of the documents in $C_q$, sorted in descending order based on their DAVINCI scores, representing the most suitable citations for a context.

### 3.2 Rethinking the Evaluation Protocol: An Inductive Setting

A central contribution of our work is to address a fundamental yet often overlooked limitation in the standard evaluation protocol for citation recommendation. Traditionally, models are evaluated in a transductive setting. In this setup, the corpus of candidate documents is often constructed from the union of training, validation and test sets, and also the unparsable documents. While this does not lead to direct data leakage (i.e., using test labels for training), it creates an artificial evaluation landscape. Specifically, the ground truth citation for a given test query itself may be another document within the test set. This means the system is evaluated on its ability to find connections within a static collection where the query documents themselves are pre-indexed and searchable which is a condition that never holds in a real-world application. To faithfully address this shortcoming, we define and adopt a rigorous inductive evaluation setting. The core principle of the inductive setting is to enforce a strict temporal separation between the evaluation query and the candidate corpus, mirroring the natural arrow of time in research.

Formally, let $D_{\mathrm{eval}}$ be an evaluation set (either the validation set, $D_{\mathrm{val}}$, or the test set, $D_{\mathrm{test}}$), and let $C$ be the candidate corpus available for recommendation. The inductive setting imposes two critical constraints:

1. **Disjoint Sets:** The set of evaluation documents and the candidate corpus must be strictly disjoint, as defined by:
$$D_{\mathrm{eval}} \cap C = \emptyset \tag{1}$$

2. **Temporal Consistency:** For any query document $D_q \in D_{\mathrm{eval}}$, the candidate corpus $C$ must only contain documents published strictly before $D_q$, formalised as:
$$\forall D_q \in D_{\mathrm{eval}}, \forall D_c \in C : \mathrm{date}(D_c) < \mathrm{date}(D_q) \tag{2}$$

This setup ensures that, at evaluation time, a model is tasked with recommending citations for a "newly authored" paper ($D_q$) using only the body of "existing" literature ($C$). By adopting this inductive protocol, we eliminate any artificial advantage gained from a pre-known test set and obtain a more realistic and reliable assessment of a model's true generalisation capabilities. All experiments and benchmarks presented in this paper are conducted under this stringent inductive setting to ensure a fair and meaningful comparison. We show the statistics for benchmark datasets in Table 1.

Table 1: The impact of our rigorous inductive setting. Enforcing temporal consistency corrects the inflation in corpus and evaluation sets seen in standard benchmarks, resulting in a markedly smaller and more realistic set of documents for training and inference. 'FTPR': FullTextPeerRead.

| Dataset | Transductive | | | | Inductive | | | |
| --- | --- | --- | --- | --- | --- | --- | --- | --- |
| | # Contexts | | | # Papers | # Contexts | | | # Corpus |
| | Train | Val | Test | | Train | Val | Test | |
| ACL-200 | 30,390 | 9,381 | 9,585 | 19,776 | 30,390 | 8,512 | 7,072 | 7,108 |
| FTPR | 9,363 | 492 | 6,814 | 4,837 | 9,363 | 472 | 5,918 | 3,313 |
| RefSeer | 3,521,582 | 124,911 | 126,593 | 624,957 | 3,521,582 | 117,724 | 105,411 | 580,059 |
| arXiv | 2,988,030 | 112,779 | 104,401 | 1,661,201 | 2,988,030 | 103,125 | 95,247 | 700,403 |
| ArSyTa | 8,030,837 | 124,188 | 124,189 | 474,341 | 8,030,837 | 123,515 | 122,989 | 412,127 |

### 3.3 Profiler: A Non-Learnable First-Stage Retrieval

The first stage of our system is the Profiler, a novel retrieval module designed to overcome the computational bottlenecks inherent in current state-of-the-art citation recommendation systems. Its design philosophy is rooted in decoupling the expensive process of representation enrichment of documents from the online query task. A key technical merit of Profiler is that it is entirely a non-learnable module. It operates as a principled, static transformation of the citation network, making it exceptionally fast and scalable. The name 'Profiler' reflects its core function: to compute a rich *public profile* for every document. We posit that a paper's relevance is a function of both its intrinsic content and its perceived identity within the scholarly network, i.e., an identity shaped by its citing papers and the contexts of those citations.

**Profiled Document Representations: A Static Enrichment Process** The Profiler's first task is a one-off, offline pre-processing step: transforming the entire corpus into a *profiled citation network*. For every document $D_i \in C$, we begin by initialising its base vector representation, $v_i \in \mathbb{R}^{d_{\text{ENC}_1}}$, using a small pre-trained language model encoder, $\text{ENC}_1(\cdot)$. We use specter2_base (Singh et al., 2022) as the encoder due to its better performance observed with citation, as reported in Goyal et al. (2024). To construct the profile of $D_i$, we augment this base representation with signals from its inward ego network, $\mathcal{N}_{in}(D_i)$, which is the set of documents that cite $D_i$. For each citing document $D_j \in \mathcal{N}_{in}(D_i)$, we extract two distinct signals: the representation of the citing paper's content, $v_j$, and the representation of the specific citation context snippet, $v_{s_{ji}}$, in which the citation is made.

The final profiled representation, $\hat{v}_i$, for document $D_i$ is a static fusion of these signals. It is formally defined as the sum of its base representation and the aggregated vectors from its citing neighbours:

$$\hat{v}_i = v_i + \frac{1}{|\mathcal{N}_{in}(D_i)|} \sum_{D_j \in \mathcal{N}_{in}(D_i)} (\alpha \cdot v_{s_{ji}} + \beta \cdot v_j) \tag{3}$$

Here, $\alpha \in [0,1]$ and $\beta \in [0,1]$ are non-learnable hyperparameters, where $\alpha + \beta = 1$. This inherently robust design formulation provides a crucial regularising effect. For a very recent paper with no citations ($|\mathcal{N}_{in}(D_i)| = 0$), the profiled representation naturally defaults to its base semantic vector, $v_i$, directly tackling the cold start problem in a principled and graceful manner. Thus, the retrieval stage becomes purely content-based for such papers. Importantly, this is not a special fallback mechanism but a natural consequence of our designed formulation. As inbound citations accumulate over time, the representation is progressively enriched through degree-normalised aggregation. Therefore, Profiler provides monotonic refinement: it never degrades below semantic retrieval performance and strictly improves representation quality as structural

evidence becomes available. This design ensures robustness under the inductive protocol, where newly published papers may initially lack citation signals.

Concurrently, averaging mechanism ensures that the profiles of highly cited papers are not unduly skewed, while effectively modelling papers from emerging fields with sparse citations and interdisciplinary work with diverse citation patterns. Hence, highly cited papers are not inherently up-weighted; only the semantic content of their citing contexts contributes to enrichment. While Profiler leverages citation structure, it deliberately excludes citation counts, venue prestige, and temporal weighting, thus mitigating the popularity bias.

Profiler can be interpreted as a deterministic one-step ratio-composite feature propagation operator over the citation graph. Specifically, Eq. 3 performs degree-normalised aggregation over the inward ego network, analogous to a fixed-weight graph convolution without learned parameters. Unlike GCN-style propagation, it introduces no learned filters, no degree amplification, and no spectral assumptions, thereby avoiding over-smoothing and hub amplification. This interpretation situates Profiler within graph representation learning as a parameter-free, content-aware smoothing operator that captures how a paper is semantically contextualised by its citing literature.

**Query Formulation and Efficient Cosine Similarity Search**   For an incoming query $q = (S_q, M_q)$, we formulate a composite query representation, $v_q$, using a similar curation strategy:

$$
\begin{aligned}
v_q &= \gamma \cdot \mathrm{ENC}_1(S_q) + \delta \cdot \mathrm{ENC}_1(M_q) \\
&= \gamma \cdot \mathrm{ENC}_1(S_q) + \delta \cdot \mathrm{ENC}_1(T_q \oplus A_q)
\end{aligned}
\tag{4}
$$

where $\oplus$ denotes textual concatenation, and $\gamma \in [0, 1]$ and $\delta \in [0, 1]$ are non-learnable hyperparameters constrained by $\gamma + \delta = 1$. With the entire corpus of profiled vectors ($\hat{v}_i$) pre-computed and indexed, the retrieval stage is reduced to a remarkably efficient similarity search. We employ cosine similarity to score the relevance of each candidate document against the query:

$$
\mathrm{Score}(q, D_i) = \mathrm{cosine}(v_q, \hat{v}_i)
\tag{5}
$$

The resulting similarity scores are not only used to rank the initial candidate list $C_q$ but are also passed directly to DAVINCI as a valuable set of confidence scores, $\{s_i\}$.

**Hyperparameter Selection**   The values for non-learnable hyperparameters $(\alpha, \beta, \gamma, \delta)$ are determined empirically via a systematic sweep analysis on the validation sets of two of our smaller datasets. Crucially, as shown in Figure 1, the optimal set of values identified from this constrained analysis ($\alpha = 0.8$, $\delta = 0.3$) is then applied universally across all larger datasets without further tuning to ensure generalisation. Our analysis revealed that a specific ratio, i.e., the one that moderately prioritises the local context signal ($\alpha = 0.8$, $\gamma = 0.7$) over the global document topic ($\beta = 0.2$, $\delta = 0.3$) yields consistently strong performance. This finding underscores that the effectiveness of Profiler doesn't lie in dataset-specific tuning, but in its ability to capture a fundamental and generalisable structural property of scholarly networks.

### 3.4   The DAVINCI Reranking Architecture

The efficacy of second-stage reranker is fundamentally constrained by its ability to enrich the semantic information of the query and the candidates obtained from the first-stage retrieval. We posit that state-of-the-art performance hinges not merely on the power of semantic encoding, but also on the confidence priors. Moreover, it depends on the sophistication of fusion mechanism that reconciles these modalities.

To this end, we introduce **DAVINCI (Discriminative & Adaptive Vector-gated Integration of Network Confidence & Information)**. It is founded on two core concepts: (i) a principled, non-linear transformation to refine the low information signal from the retrieval stage, and (ii) a novel fashion that creates a soft masking mechanism to achieve a dynamic and fine-grained fusion of signals. Finally, the reranker is optimised end-to-end using contrastive learning.

**From Degenerate Scores to Discriminative Priors**   A prerequisite for effective fusion is the availability of well-informed input signals. Raw cardinal scores from dense retrievers often exhibit severe score

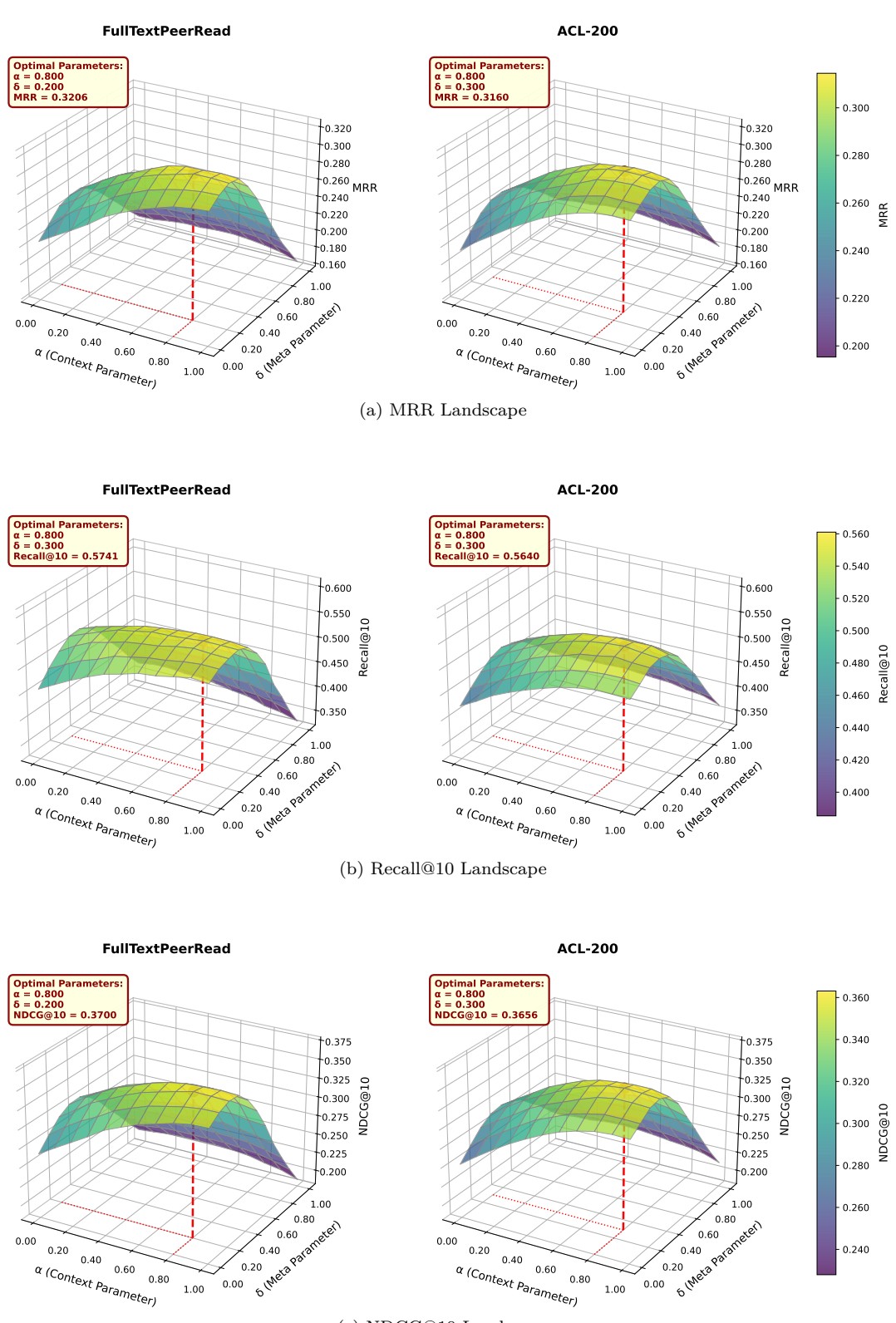

(a) MRR Landscape

(b) Recall@10 Landscape

(c) NDCG@10 Landscape

Figure 1: Navigating the performance landscape of the public profile enrichment on the FullTextPeerRead and ACL-200 validation sets. Each plot shows a different evaluation metric: (a) MRR, (b) Recall@10, and (c) NDCG@10.

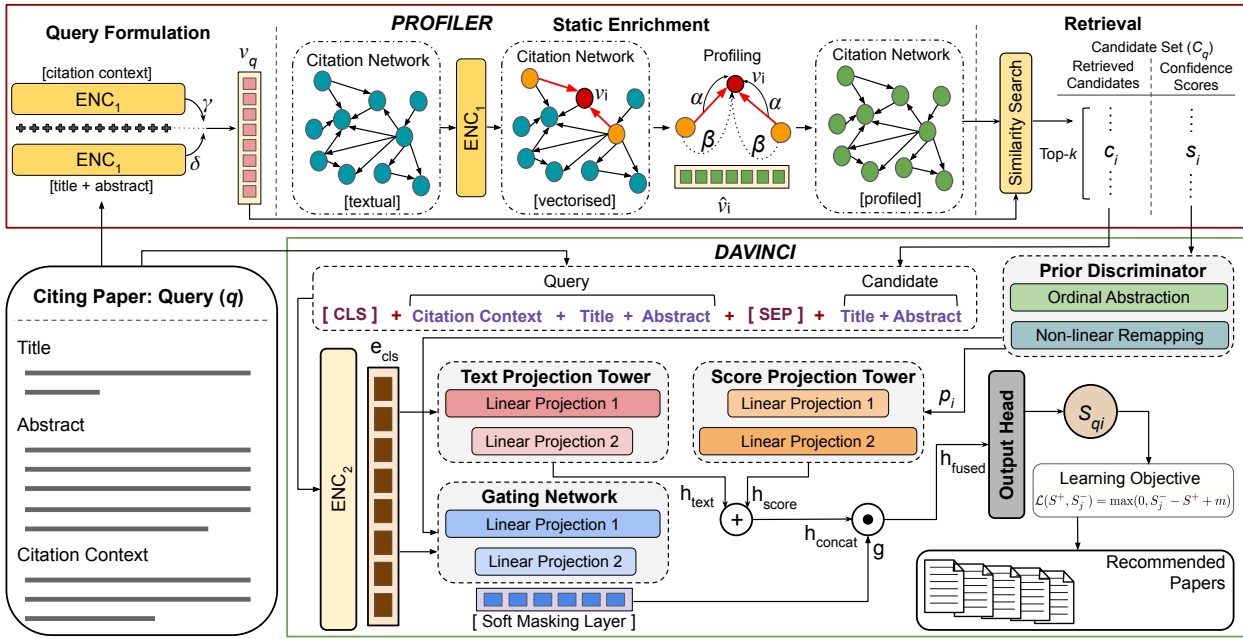

Figure 2: The architecture of our two-stage citation recommendation system. (1) The non-learnable **Profiler** performs a scalable retrieval by matching the query against a corpus of documents enriched with their *public profile.* (2) **DAVINCI** reranks the retrieved candidates using a vector-gated mechanism to integrate the discriminative retrieval priors with deep semantic features to produce a final ranked list of recommended papers for citation.

compression, providing a low-information signal with poor discriminative capacity. We therefore introduce a deterministic preprocessing block to transform this signal into a robust retrieval prior. (i) **Ordinal Abstraction**: We obtain a 1-indexed rank list $\{r_i\}$ from the list of cardinal scores $\{s_i\}$. For any ground-truth candidate not found in the profiler's output (e.g., an oracle-provided positive injected for training), we assign a default rank of $k + 1$, where $k = |C_q|$. (ii) **Non-Linear Remapping**: The resulting integer ranks, while robust, are both linearly spaced and numerically large, and thus fails to capture the power-law distribution of relevance in ranked lists. These large integer values can be problematic for gradient-based optimisation, potentially leading to unstable training or exploding gradients. To address both issues simultaneously, we apply a non-linear exponential decay function to map the rank $r_i$ to final transformed prior $p_i$:

$$p_i = \lambda^{r_i} \tag{6}$$

where $\lambda \in (0, 1)$ is a decay-rate hyperparameter, empirically set to 0.95. This transformation yields a geometrically spaced, continuous prior that more accurately models the steep non-linear decay of relevance probability. This transformed prior $p_i$ serves as the definitive retrieval signal for all subsequent model components.

**Adaptive Gated Fusion**  The DAVINCI design is engineered to leverage the discriminative confidence prior $p_i$ and fuse it intelligently with the raw semantic information. The semantic information is obtained by textually concatenating query text with the candidate text using a [SEP] token and encoding it using a small pretrained language model, $ENC_2(\cdot)$. We use SciBERT (Beltagy et al., 2019) as the encoder due to its better performance observed with non-graph fusion techniques, as reported in Goyal et al. (2024). We extract the [CLS] token's final hidden state, $e_{cls} \in \mathbb{R}^{d_{ENC_2}}$, as the raw semantic representation. To enable fusion, the heterogeneous inputs are first mapped into a common $d_h$-dimensional latent space via two independent Multi-Layer Perceptron (MLP) towers representing modality specific projection networks:

- **Text Projection Tower (MLP$_{\text{text}}$):** Learns a non-linear mapping $f_{\text{text}} : \mathbb{R}^{d_{\text{ENC}_2}} \to \mathbb{R}^{d_h}$, yielding a task-specific text representation $h_{\text{text}}$.

- **Score Projection Tower (MLP$_{\text{score}}$):** Learns a mapping $f_{\text{score}} : \mathbb{R} \to \mathbb{R}^{d_h}$, vectorising the scalar prior $p_i$ into a dense score representation $h_{\text{score}}$.

To obtain the final processed semantic information, projected representations are concatenated as:

$$h_{\text{concat}} = [h_{\text{text}}; h_{\text{score}}] \in \mathbb{R}^{2d_h} \tag{7}$$

A separate **Gating Network**, **MLP$_{\text{gate}}$**, then computes a vector-valued gate $g$. This network is conditioned on the original input signals ($e_{\text{cls}}$ and $p_i$) to form an unbiased assessment of the raw evidence:

$$g = \sigma(\text{MLP}_{\text{gate}}([e_{\text{cls}}; p_i])) \in \mathbb{R}^{2d_h} \tag{8}$$

Here, $\sigma$ is the element-wise sigmoid function, which constrains each element of the gating vector $g$ to the range $(0, 1)$. Each element $g_j$ can be interpreted as a learned throughput coefficient for the $j$-th feature. The final fusion is executed via the Hadamard product ($\odot$), which applies the gate $g$ as a **per-dimension soft mask**:

$$h_{\text{fused}} = g \odot h_{\text{concat}} \tag{9}$$

This operation constitutes a form of *element-wise feature modulation*, providing a degree of representational flexibility unattainable with scalar fusion methods. The adaptively fused vector, $h_{\text{fused}}$, is passed to a dedicated **Output Head**, a final MLP (**MLP$_{\text{out}}$**), which maps the $2d_h$-dimensional representation to a single logit. A final sigmoid activation produces the final reranked DAVINCI score $S_{qi} = f_{\text{DAVINCI}}(q, c_i, s_i)$ as shown below

$$S_{qi} = \sigma(\text{MLP}_{\text{out}}(h_{\text{fused}})) \in (0, 1) \tag{10}$$

This score $S_{qi}$ represents the system's final confidence that candidate document $D_i$ is a relevant citation for query $q$.

**Learning Objective: Direct Optimisation of Ranking.** To align model's training with its downstream evaluation, we use a loss function that directly optimises the relative ordering of candidates. The training process is structured around queries and their associated sets of $k$ retrieved document candidates, which are labeled as positive ($c^+$) or negative ($c^-$) based on ground-truth relevance. To construct robust training instances and expose the model to a diverse set of negative signals, we adopt a negative sampling strategy. For a positive candidate $c^+$ associated with a query $q$, we compare it with randomly sampled $n$ negative candidates, denoted as $\{c_1^-, c_2^-, \ldots, c_n^-\}$, from the pool of $k$ retrieved candidates for the query. This process yields $n$ distinct training triplets for a positive example. For each triplet $(q, c^+, c_j^-)$, the model computes the respective scores, $S^+$ and $S_j^-$. We then optimise the model using the margin-based triplet loss, applied individually to each pair:

$$\mathcal{L}(S^+, S_j^-) = \max(0, S_j^- - S^+ + m) \tag{11}$$

where $m \in (0, 1)$ is a margin hyperparameter. The total loss for a positive sample $c^+$ is the average sum of losses computed over these $n$ sampled negatives: $\frac{1}{n} \sum_{j=1}^{n} \mathcal{L}(S^+, S_j^-)$. This objective function directly penalises incorrect rank-ordering across a varied subset of competitors, forcing the model to learn a scoring function that produces a well-separated ranking of candidates. The entire architecture is shown in Fig. 2.

## 4 Experiments and Results

### 4.1 Implementation Details

This section details the experimental protocol designed to rigorously evaluate our proposed work. We conduct all experiments and benchmark under the realistic inductive setting. Our experimental pipeline is designed to reflect the distinct computational profiles of retrieval and reranking. The coarse-grained retrieval stages for all systems are executed on NVIDIA A100 DGX clusters. The more computationally intensive, fine-grained

Table 2: Our retrieval module (**Profiler**) consistently outperforms the SOTA baselines on all datasets across metrics and also with respect to the computational timing. Pref+Enr refers to sequential combination of Prefetcher followed by Enricher, leading to higher Recall@300 and NDCG@300 while keeping the same metric values for K=10 and K=50 as per its enrichment principle. Experiments are run on NVIDIA A100 DGX.

| Model | Comp. Time | MRR | Recall@K | | | NDCG@K | | |
|---|---|---|---|---|---|---|---|---|
| | | | 10 | 50 | 300 | 10 | 50 | 300 |
| **ACL-200** | | | | | | | | |
| Prefetcher | 56.22m | 21.14 | 40.33 | 65.37 | 86.98 | 24.57 | 30.11 | 33.30 |
| Pref+Enr | 64.43m | 21.16 | 40.33 | 65.37 | 88.93 | 24.57 | 30.11 | 33.48 |
| **Profiler** | **2.52m** | **30.17** | **53.79** | **74.63** | **89.58** | **34.88** | **39.57** | **41.78** |
| **FullTextPeerRead** | | | | | | | | |
| Prefetcher | 45.61m | 21.73 | 39.17 | 63.43 | 87.16 | 24.78 | 30.15 | 33.63 |
| Pref+Enr | 49.20m | 21.76 | 39.17 | 63.43 | 88.40 | 24.78 | 30.15 | 33.97 |
| **Profiler** | **1.12m** | **31.62** | **57.23** | **82.05** | **96.27** | **36.62** | **42.23** | **44.35** |
| **Refseer** | | | | | | | | |
| Prefetcher | 99.17h | 11.88 | 22.72 | 41.88 | 66.76 | 13.56 | 17.77 | 21.39 |
| Pref+Enr | 101.43h | 11.92 | 22.72 | 41.88 | 69.91 | 13.56 | 17.77 | 21.88 |
| **Profiler** | **3.10h** | **16.65** | **32.18** | **52.46** | **72.17** | **19.40** | **23.91** | **26.80** |
| **arXiv** | | | | | | | | |
| Prefetcher | 84.31h | 13.78 | 27.09 | 48.83 | 74.16 | 15.94 | 20.73 | 24.43 |
| Pref+Enr | 85.94h | 13.80 | 27.09 | 48.83 | 76.24 | 15.94 | 20.73 | 24.96 |
| **Profiler** | **2.72h** | **16.61** | **33.41** | **55.95** | **76.61** | **19.56** | **24.57** | **27.61** |
| **ArSyTa** | | | | | | | | |
| Prefetcher | 225.88h | 7.89 | 15.52 | 31.08 | 56.00 | 8.96 | 12.36 | 15.95 |
| Pref+Enr | 236.14h | 7.94 | 15.52 | 31.08 | 66.59 | 8.96 | 12.36 | 17.31 |
| **Profiler** | **7.26h** | **13.01** | **26.36** | **47.46** | **69.35** | **15.17** | **19.84** | **23.04** |

reranking stages utilise NVIDIA H200 DGX systems to ensure efficient processing. Given the substantial scale of the corpora and datasets, conducting multiple full training runs is computationally prohibitive. To ensure the robustness of our findings, we first perform a stability analysis. We conduct three training trials on representative subsets of the training data and observed minimal variance in performance, confirming the numerical stability of our training procedure. Consequently, the final results reported in all tables are from a single, comprehensive run on the full-scale datasets. The hyperparameters used for training DAVINVI are: Learning_rate = 1e-5; L2_weight = 0.01; Dropout = 0.3; Optimiser = AdamW; Adam_epsilon = 1e-6; Loss_margin = 0.1 and Batch_size = 16. To provide a multi-faceted assessment of ranking performance, we employ a suite of standard information retrieval metrics (%), namely, Mean Reciprocal Rank (MRR), Recall@K, and NDCG@K.

## 4.2 Results: First Stage Retrieval

We compare the results of our Profiler with the current state-of-the-art Prefetcher (Gu et al., 2022) and the sequential combination of Prefetcher followed by Enricher (Goyal et al., 2024). Prefetcher operates on a hierarchical attention based text encoding to obtain a list of first stage retrieved candidates for citation recommendation. Enricher ingests top 100 candidates from this prefetched list and models their symbiotic relationship embedded in the citation network to curate an enriched list of retrieved candidates, thus yielding a significantly higher Recall@300. In Table 2, results show that the non-learnable and scalable nature of Profiler makes it highly computationally efficient in reducing the retrieval time by 32.52x and 43.92x on

Table 3: Performance comparison showing our end-to-end citation recommendation system (**Ours**) consistently outperforming all baselines.

| Model | MRR | Recall@K | | | NDCG@K | | |
|---|---|---|---|---|---|---|---|
| | | **5** | **10** | **20** | **5** | **10** | **20** |
| **ACL-200** | | | | | | | |
| BM25 | 10.53 | 15.45 | 20.82 | 26.71 | 10.71 | 12.44 | 13.92 |
| SciNCL | 15.41 | 21.39 | 30.04 | 39.76 | 15.01 | 17.79 | 20.24 |
| HAtten | 45.53 | 58.93 | 68.24 | 75.78 | 47.32 | 50.34 | 52.25 |
| SymTax | 46.98 | 60.20 | 69.47 | 76.83 | 48.73 | 51.75 | 53.62 |
| **Ours** | **50.31** | **64.10** | **73.08** | **80.20** | **52.30** | **55.22** | **57.03** |
| **FullTextPeerRead** | | | | | | | |
| BM25 | 16.60 | 24.50 | 31.15 | 38.23 | 17.27 | 19.42 | 21.23 |
| SciNCL | 17.80 | 25.31 | 35.43 | 46.48 | 17.53 | 20.77 | 23.57 |
| HAtten | 55.03 | 68.60 | 75.58 | 80.62 | 57.33 | 59.60 | 60.88 |
| SymTax | 56.63 | 69.94 | 76.92 | 82.29 | 58.84 | 61.11 | 62.47 |
| **Ours** | **59.68** | **74.41** | **82.17** | **87.42** | **62.16** | **64.68** | **66.02** |
| **Refseer** | | | | | | | |
| BM25 | 10.85 | 15.31 | 19.71 | 24.50 | 11.11 | 12.52 | 13.73 |
| SciNCL | 7.17 | 10.02 | 14.68 | 20.46 | 6.74 | 8.23 | 9.69 |
| HAtten | 30.64 | 39.41 | 45.78 | 51.41 | 32.01 | 33.72 | 34.98 |
| SymTax | 31.80 | 40.61 | 47.24 | 53.25 | 32.79 | 34.94 | 36.46 |
| **Ours** | **32.57** | **42.19** | **49.52** | **56.37** | **33.62** | **36.00** | **37.73** |
| **arXiv** | | | | | | | |
| BM25 | 10.28 | 14.64 | 19.04 | 23.89 | 10.50 | 11.93 | 13.15 |
| SciNCL | 9.22 | 13.06 | 18.37 | 24.89 | 8.91 | 10.61 | 12.25 |
| HAtten | 28.13 | 37.01 | 45.06 | 52.32 | 28.86 | 31.36 | 32.37 |
| SymTax | 29.02 | 38.46 | 46.78 | 54.97 | 29.80 | 32.49 | 34.56 |
| **Ours** | **30.46** | **40.86** | **49.89** | **58.50** | **31.38** | **34.31** | **36.49** |
| **ArSyTa** | | | | | | | |
| BM25 | 9.24 | 13.39 | 17.52 | 22.14 | 9.46 | 10.79 | 11.96 |
| SciNCL | 8.16 | 11.25 | 15.71 | 21.08 | 7.85 | 9.28 | 10.64 |
| HAtten | 19.92 | 27.70 | 34.90 | 42.25 | 20.50 | 22.83 | 24.69 |
| SymTax | 22.00 | 30.16 | 38.06 | 46.03 | 22.49 | 25.05 | 27.07 |
| **Ours** | **24.01** | **33.74** | **42.83** | **51.56** | **24.73** | **27.67** | **29.89** |

the largest dataset (ArSyTa) and the smallest dataset (FullTextPeerRead), respectively. Results also show Profiler's merit to retrieve better candidates by increasing the MRR by 63.85% and 45.3% on ArSyTa and FullTextPeerRead, respectively.

## 4.3  Results: End-to-End System

We evaluate our complete system with other standard baselines in Table 3 as detailed in our experimental setup. We outperform the SOTA citation recommendation systems and establish a new state-of-the-art on all datasets across all metrics. All the results are consistent with the inference reported in Goyal et al. (2024) with ArSyTa and arXiv being the toughest benchmarks owing to their broad spectrum of scientific papers.

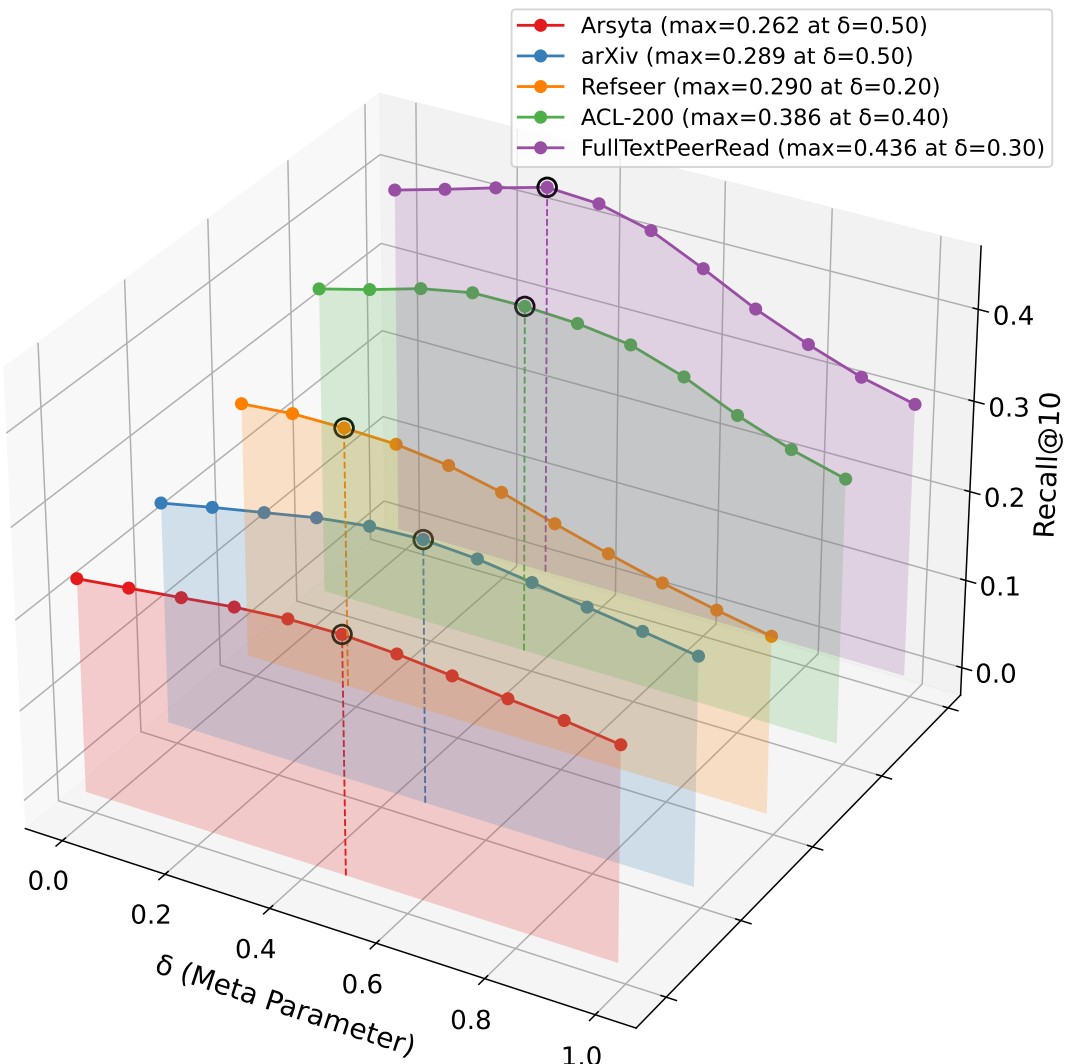

Figure 3: The performance variation for varied query composition when the profiling is disabled.

## 5    Analysis

To dissect the contributions of our core design choices, we conduct a series of targeted **ablation** studies on both the Profiler and the DAVINCI reranker. These analyses are designed to validate our architectural hypotheses and quantify the impact of each novel component. Additionally, we present both the **quantitative analysis** and the **qualitative analysis**.

### 5.1    Ablation Analysis: Profiler

We perform two key analyses to validate the efficacy of the public profile concept and its implementation in the Profiler. In Figure 1, we visualise and navigate the landscape of public profile corresponding to FullTextPeerRead and ACL-200 datasets for MRR, Recall@10 and NDCG@10, clearly depicting the entire spectrum of public profile. To measure the performance gain enabled by profiling, we conduct an ablation where the profile enrichment is turned off (i.e., setting $\alpha = 0$ and $\beta = 0$ in Equation 3, so $\hat{v}_i = v_i$). As shown in Fig. 3, we observe a significant degradation in retrieval performance for all datasets across varied query compositions (i.e., different $\gamma, \delta$ values). Moreover, we observe that large and tough datasets are relatively more robust to varied query compositions in this case. We also show the scores for Recall@10 on all datasets

Table 4: Performance comparison w.r.t. Recall@10 on all datasets with profiling enabled versus the maximum attainable scores possible for any query composition when the profiling is disabled.

|  | ACL-200 | FTPR | Refseer | arXiv | ArSyTa |
|---|---|---|---|---|---|
| w/ Profiling | 53.7 | 57.2 | 32.1 | 33.4 | 26.3 |
| max attainable w/o Profiling | 38.6 | 43.6 | 29.0 | 28.9 | 26.2 |

Table 5: Ablation analysis showing the impact of our design choices w.r.t. **our** complete system, namely, **A1** (Semantics Only), **A2** (Turned-off Discriminator), **A3** (Softmax Normalisation), and **A4** (Scalar Gating).

| Model | MRR | Recall@K | | | NDCG@K | | |
|---|---|---|---|---|---|---|---|
| | | 5 | 10 | 20 | 5 | 10 | 20 |
| **ACL-200** | | | | | | | |
| **Ours** | **50.31** | **64.10** | **73.08** | **80.20** | **52.30** | **55.22** | **57.03** |
| A1 | 48.42 | 62.42 | 71.32 | 78.38 | 50.44 | 53.34 | 55.13 |
| A2 | 48.30 | 61.85 | 70.75 | 77.81 | 50.20 | 53.10 | 54.89 |
| A3 | 49.46 | 62.67 | 71.83 | 78.49 | 51.27 | 54.26 | 55.96 |
| A4 | 45.16 | 57.66 | 66.05 | 72.99 | 46.81 | 49.54 | 51.30 |
| **FullTextPeerRead** | | | | | | | |
| **Ours** | **59.68** | **74.41** | **82.17** | **87.42** | **62.16** | **64.48** | **66.02** |
| A1 | 58.08 | 72.88 | 80.30 | 86.19 | 60.56 | 62.96 | 64.46 |
| A2 | 58.20 | 72.78 | 80.20 | 86.26 | 60.61 | 63.03 | 64.56 |
| A3 | 58.49 | 72.90 | 80.88 | 86.23 | 60.83 | 63.42 | 64.78 |
| A4 | 53.58 | 68.04 | 75.90 | 82.22 | 55.90 | 58.46 | 60.08 |

with profiling enabled versus the maximum attainable scores possible for any query composition when the profiling is disabled in Table 4. This directly confirms that profiling is not merely a hypothetical construct but a vital signal for effective first-stage retrieval.

## 5.2 Ablation Analysis: DAVINCI

To isolate the contribution of each component within DAVINCI, we conduct four ablation studies, systematically deconstructing the full model. The results for these ablations on the FullTextPeerRead and ACL-200 datasets are presented in Table 5, and are described as follows (1) **Semantics Only:** We discard the use of network confidence scores. This experiment is designed to quantify the value of integrating the Profiler's retrieval confidence into the reranking stage. (2) **Turned-off Discriminator:** We bypass our signal refining process (ordinal abstraction and exponential remapping) and instead feed the raw, untransformed confidence scores from the Profiler to testify the necessity of our proposed transformation for handling low-information retrieval signals. (3) **Softmax Normalisation:** We replace our discriminative transformation with a standard softmax function applied to the retrieval scores of the top-k candidates. This provides a direct comparison of our principled remapping scheme against a common baseline for score normalisation. (4) **Scalar Gating:** We replace the vector-gating mechanism with scalar gating of semantic information controlled by discriminative prior. This experiment directly measures the performance gain attributable to our fine-grained, per-dimension adaptive fusion policy.

## 5.3 Quantitative Analysis

To analyse the sensitivity of our reranker to the candidate pool size, we present a quantitative analysis varying the number of candidates (k) to be reranked. While the main experiments in this paper are conducted with k=300, Table 6 details the performance variation at different values of k. The results reveal two distinct trends: on smaller datasets, performance scales positively with k; however, on larger datasets, performance

Table 6: Analysis showing the impact of number of candidates (**k**) on reranking performance. We found the value of 300 as an overall better choice for the final reranking performance with respect to the metrics and the computational overhead.

| k | MRR | Recall@K | | | NDCG@K | | |
|---|---|---|---|---|---|---|---|
| | | **5** | **10** | **20** | **5** | **10** | **20** |
| **ACL-200** | | | | | | | |
| 50 | 46.97 | 59.62 | 67.02 | 71.75 | 49.04 | 51.46 | 52.67 |
| 100 | 48.92 | 62.08 | 70.36 | 76.46 | 50.92 | 53.62 | 55.17 |
| 300 | 50.31 | 64.10 | 73.08 | 80.20 | 52.30 | 55.22 | 57.03 |
| 1000 | **50.92** | **64.86** | **74.31** | **81.73** | **52.84** | **55.91** | **57.79** |
| **FullTextPeerRead** | | | | | | | |
| 50 | 55.03 | 68.60 | 75.14 | 79.35 | 57.48 | 59.61 | 60.68 |
| 100 | 57.69 | 72.01 | 79.25 | 83.87 | 60.19 | 62.54 | 63.72 |
| 300 | 59.68 | 74.41 | 82.17 | 87.42 | 62.16 | 64.68 | 66.02 |
| 1000 | **60.20** | **75.14** | **82.84** | **88.43** | **62.71** | **65.22** | **66.64** |
| **Refseer** | | | | | | | |
| 50 | 28.78 | 37.47 | 43.61 | 48.73 | 29.96 | 31.95 | 33.25 |
| 100 | 30.60 | 39.77 | 46.55 | 52.66 | 31.70 | 33.90 | 35.45 |
| 300 | 32.57 | **42.19** | **49.52** | **56.37** | 33.62 | **36.00** | **37.73** |
| 1000 | **32.74** | 42.01 | 49.11 | 55.89 | **33.68** | 35.98 | 37.70 |
| **arXiv** | | | | | | | |
| 50 | 27.24 | 37.12 | 45.01 | 51.55 | 28.44 | 31.00 | 32.66 |
| 100 | 28.85 | 39.02 | 47.65 | 55.33 | 29.89 | 32.69 | 34.64 |
| 300 | **30.46** | **40.86** | **49.89** | **58.50** | **31.38** | **34.31** | **36.49** |
| 1000 | 30.06 | 39.82 | 48.62 | 56.97 | 30.81 | 33.66 | 35.78 |
| **ArSyTa** | | | | | | | |
| 50 | 21.51 | 30.52 | 37.73 | 43.67 | 22.60 | 24.94 | 26.45 |
| 100 | 22.96 | 32.47 | 40.60 | 47.84 | 23.93 | 26.57 | 28.40 |
| 300 | **24.01** | **33.74** | **42.83** | **51.56** | **34.73** | **27.67** | **29.89** |
| 1000 | 20.74 | 29.24 | 38.37 | 47.98 | 21.02 | 23.96 | 26.39 |

peaks around k=300 and subsequently degrades. This degradation suggests that processing too many low-quality candidates introduces noise that can harm the reranker's precision. Given that computational cost also grows linearly with k, this analysis confirms that k=300 represents an optimal trade-off, maximising performance while avoiding the dual penalties of increased noise and computational overhead.

## 5.4 Qualitative Analysis

To complement our quantitative results and provide deeper insight into the mechanisms driving our model's performance, we conduct a qualitative case study. By manually inspecting the recommendations for a representative query, we can better understand how our system compares with the state-of-the-art citation recommendation systems, as shown in Table 7. We select a query paper from our test set whose topic is nuanced and requires a deep understanding of the semantics. The SOTA models demonstrate a classic failure mode of relying on broad and superficial topic matching. They correctly identify the general topic of 'Machine Translation' but completely misses the critical and specific usage of the term 'MERT'. Instead they focus on another term 'Moses' from both the citation context and the query abstract, and use these two signals to recommend from the candidate pool. On the other hand, our system also identify the same

Table 7: Case study of citation recommendations for a sample from the ACL-200 dataset. The table contrasts the top-10 predictions from SOTA baseline models against our system, with **ground-truth** citation highlighted in **bold** to illustrate our model's improved relevance. We can see that our model is successfully able to predict the correct citation by checking the abbrevation 'MERT' against the titles of the available candidates whereas the other systems just focus on the term ('Moses') in the abstract of the citing paper and the citation context, and use it for checking. # denotes the rank of the recommended citations.

**Citation Context:-** "lation, phrases are extracted from this synthetic corpus and added as a separate phrase table to the combined system (CH1). The relative importance of this phrase table is estimated in standard MERT ( TARGETCIT) . The final translation of the test set is produced by Moses (enriched with this additional phrase table) and additionally post-processed by Depfix. Note that all components of this combination have d"

**Query Title:-** What a Transfer-Based System Brings to the Combination with PBMT.

**Query Abstract:-**We present a thorough analysis of a combination of a statistical and a transferbased system for English→Czech translation, Moses and TectoMT. We describe several techniques for inspecting such a system combination which are based both on automatic and manual evaluation. While TectoMT often produces bad translations, Moses is still able to select the good parts of them. In many cases, TectoMT provides useful novel translations which are otherwise simply unavailable to the statistical component, despite the very large training data. Our analyses confirm the expected behaviour that TectoMT helps with preserving grammatical agreements and valency requirements, but that it also improves a very diverse set of other phenomena. Interestingly, including the outputs of the transfer-based system in the phrase-based search seems to have a positive effect on the search space. Overall, we find that the components of this combination are complementary and the final system produces significantly better translations than either component by itself.

| # | HAtten recommendation | SymTax recommendation | Ours recommendation |
|---|---|---|---|
| 1 | Moses: Open Source Toolkit for Statistical Machine Translation | Moses: Open Source Toolkit for Statistical Machine Translation | **Minimum Error Rate Training in Statistical Machine Translation** |
| 2 | Combining Multi-Engine Translations with Moses | Findings of the 2012 Workshop on Statistical Machine Translation | Statistical Phrase-Based Translation |
| 3 | SMT and SPE Machine Translation Systems for WMT'09 | A STATISTICAL APPROACH TO MACHINE TRANSLATION | Moses: Open Source Toolkit for Statistical Machine Translation |
| 4 | MANY: Open Source MT System Combination at WMT'10 | Combining Multi-Engine Translations with Moses | Findings of the 2012 Workshop on Statistical Machine Translation |
| 5 | Edinburgh's Machine Translation Systems for European Language Pairs | Phrasetable Smoothing for Statistical Machine Translation | Improved Statistical Alignment Models |
| 6 | Toward Using Morphology in French-English Phrase-based SMT | **Minimum Error Rate Training in Statistical Machine Translation** | A STATISTICAL APPROACH TO MACHINE TRANSLATION |
| 7 | Parallel Implementations of Word Alignment Tool | Training Phrase Translation Models with Leaving-One-Out | Improvements in Phrase-Based Statistical Machine Translation |
| 8 | Improved Alignment Models for Statistical Machine Translation | SMT and SPE Machine Translation Systems for WMT'09 | Combining Multi-Engine Translations with Moses |
| 9 | Investigations on Translation Model Adaptation Using Monolingual Data | Statistical Phrase-Based Translation | Hierarchical Phrase-Based Translation |
| 10 | A STATISTICAL APPROACH TO MACHINE TRANSLATION | MANY: Open Source MT System Combination at WMT'10 | Phrasetable Smoothing for Statistical Machine Translation |

general topic of 'Machine Translation' but intelligently picking up the abbreviated term 'MERT' and using it effectively for recommending from the retrieved candidate set by comparing it with their titles.

# 6 Comparison with Massive-Scale Rerankers

We conduct an experiment to answer a critical question: Can a compact, purpose-built reranker like DAVINCI outperform general-purpose reranking models with orders of magnitude more parameters that rely primarily on scale and broad pretraining? We evaluate against the current state-of-the-art reranking models, including the latest Qwen3-Reranker-8B (Zhang et al., 2025) and bge-reranker-v2-minicpm-40 having 2.72B parameters (Chen et al., 2024; Li et al., 2023). In contrast, our DAVINCI model is exceptionally lightweight, comprising only 110M parameters. To ensure a fair comparison, we standardise the retrieval stage for all models: each reranker is provided with the exact same list of candidate documents retrieved by our Profiler module, and we evaluate the performance on same test sets used for DAVINCI. Due to the immense size of these rerankers and their general-purpose pre-training, we employ instruction-aware prompt-

Table 8: Performance of DAVINCI (110M) vs. massive-scale rerankers. 'R': Reranker; 'm': minicpm. Despite being up to 70x smaller, our specialised model substantially outperforms general-purpose models like the state-of-the-art Qwen3-Reranker-8B across all datasets and metrics, demonstrating the merit of task-specific design over raw parameter scale.

| Model | MRR | Recall@K | | | NDCG@K | | |
|---|---|---|---|---|---|---|---|
| | | 5 | 10 | 20 | 5 | 10 | 20 |
| **ACL-200** | | | | | | | |
| **DAVINCI** | **50.31** | **64.10** | **73.08** | **80.20** | **52.30** | **55.22** | **57.03** |
| Qwen3-R-8B | 36.44 | 50.96 | 63.06 | 72.83 | 38.02 | 41.94 | 44.42 |
| bge-R-v2-m-40 | 33.52 | 45.27 | 55.23 | 64.70 | 34.55 | 37.78 | 40.17 |
| **FullTextPeerRead** | | | | | | | |
| **DAVINCI** | **59.68** | **74.41** | **82.17** | **87.42** | **62.16** | **64.68** | **66.02** |
| Qwen3-R-8B | 48.15 | 66.84 | 77.62 | 85.62 | 51.08 | 54.60 | 56.63 |
| bge-R-v2-m-40 | 41.22 | 53.71 | 63.87 | 73.16 | 42.44 | 45.75 | 48.11 |
| **Refseer** | | | | | | | |
| **DAVINCI** | **32.57** | **42.19** | **49.52** | **56.37** | **33.62** | **36.00** | **37.73** |
| Qwen3-R-8B | 24.81 | 35.39 | 44.98 | 54.04 | 25.67 | 28.79 | 31.09 |
| bge-R-v2-m-40 | 22.10 | 30.27 | 38.39 | 46.58 | 22.53 | 25.15 | 27.22 |
| **arXiv** | | | | | | | |
| **DAVINCI** | **30.46** | **40.86** | **49.89** | **58.50** | **31.38** | **34.31** | **36.49** |
| Qwen3-R-8B | 25.48 | 36.02 | 47.19 | 57.35 | 26.10 | 29.72 | 32.30 |
| bge-R-v2-m-40 | 21.70 | 29.79 | 38.10 | 46.87 | 21.99 | 24.68 | 26.89 |
| **ArSyTa** | | | | | | | |
| **DAVINCI** | **24.01** | **33.74** | **42.83** | **51.56** | **24.73** | **27.67** | **29.89** |
| Qwen3-R-8B | 22.39 | 32.44 | 40.71 | 49.33 | 23.26 | 25.95 | 28.13 |
| bge-R-v2-m-40 | 17.79 | 24.70 | 31.73 | 38.79 | 18.03 | 20.31 | 22.08 |

ing to adapt them to our specific task and datasets. Despite being up to *70x* smaller than the latest SOTA reranker, our specialised model markedly outperforms general-purpose models on all datasets, demonstrating the merit of task-specific design over raw parameter scale in an era of massive models, as shown in Table 8. Our findings demonstrate that performance gains arise from two complementary factors: (i) Task-specific architectural inductive bias, and (ii) Supervised adaptation to the citation recommendation objective.

## 6.1 Qwen Reranker Series

The Qwen model series, developed by Alibaba Cloud, represents a significant advancement in open-source language models. The rerankers from this series are specifically fine-tuned for relevance ranking tasks. Building upon the dense foundational models of the Qwen3 series, it provides a comprehensive range of reranking models in various sizes (0.6B, 4B, and 8B). This series inherits the exceptional multilingual capabilities, long-text understanding, and reasoning skills of its foundational model. The Qwen rerankers based on a powerful transformer architecture are trained on massive datasets of query-document pairs, learning to discern subtle relevance signals far beyond simple keyword matching. As instruction-tuned models, they operate as cross-encoders that expect a structured prompt. The model ingests the query and document by embedding them within a specific template that defines the task. This allows for deep, token-level interaction between the query and the document, conditioned on the explicit instruction. The model is trained to output a single logit, where a higher value indicates a higher probability of relevance. We select the Qwen reranker as it is widely regarded as a state-of-the-art, general-purpose reranker. Its strong performance across various public benchmarks makes it a formidable baseline to measure against. We use the latest and the largest

available open-source version, Qwen3-Reranker-8B having 8.19B parameters, from the Qwen series for our experiments.

## 6.2 BGE Reranker v2 (BAAI General Embedding)

The BGE model family, released by the Beijing Academy of Artificial Intelligence (BAAI), is another highly influential series of models optimised for text retrieval and ranking. The BGE-Reranker-v2 is particularly notable for its excellent performance and efficiency. The BGE reranker is also a cross-encoder based on a transformer architecture. It has been fine-tuned on a mixture of public and proprietary datasets specifically for relevance ranking. The model architecture, often based on efficient backbones like *minicpm*, is designed to deliver high performance without the prohibitive computational cost of the largest models. The *layerwise* aspect in some variants refers to advanced techniques that leverage representations from multiple transformer layers, which can enhance performance. The usage is identical to that of Qwen where it ingests a query, document pair and processes it through its transformer layers. It outputs a relevance logit, which is used to re-sort the candidates. BGE models are known for their strong performance on standardised retrieval benchmarks like the MTEB (Massive Text Embedding Benchmark). We employ the bge-reranker-v2-minicpm-layerwise having 40 layers and 2.72B parameters to provide another strong, publicly available baseline from a different lineage than Qwen. Its high ranking on public leaderboards and widespread adoption in the community make it an essential point of comparison for any new reranking model.

## 6.3 Implementation and Usage

To ensure a fair and direct comparison, we follow a consistent protocol for all baseline models. The pre-trained checkpoints for both the Qwen and BGE rerankers are loaded directly from the Hugging Face Hub. For each query-document pair, we use the specific instruction-based prompt formats recommended for Qwen and bge, respectively. For Qwen, an instruction, the query, and the candidate document text are combined into a single string template: `"<Instruct>: {instruction}\n<Query>: {query}\n<Document>: {doc}"`. For the {instruction} placeholder, we curate a clear task description as suggested in the Qwen guidelines: `"Given a citation context and citing paper information, determine if the candidate paper is relevant to be cited in this context"`. The sequences are truncated to the models' maximum input length. For bge, we follow its guidelines by choosing its recommended bge specific prompt: `"Given a query A and a passage B, determine whether the passage contains an answer to the query by providing a prediction of either 'Yes' or 'No'"`. We run the models in inference mode on our same evaluation sets. For each formatted input, we extract the raw logit output before any final activation. This logit is used directly as the relevance score for reranking. To reiterate, the same set of retrieved candidates and the same evaluation metrics used for our own system are applied to these baselines to maintain experimental consistency.

## 7 Conclusion

This document purely presents a work of research and is not about productising via developing a digital assistant. Our work presented a principled re-evaluation of the citation recommendation task, advancing the field on two fundamental fronts: the veracity of its benchmarks and the efficiency of its architectures. By instituting a rigorous inductive protocol, we first established a more faithful measure of real-world performance. Next, within this demanding framework, our proposed two-stage system, pairing a non-learnable retriever with a specialised gated reranker, set a new benchmark for both retrieval and end-to-end recommendation. The strong performance of our compact, 110M-parameter model against multi-billion parameter rerankers underscored a key finding: for specialised domains, architectural sophistication, task-aligned design choices and the integration of domain-specific knowledge are more salient drivers of success than just the raw parameter count.

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
