# OpenReview forum: "Public Profile Matters: A Scalable Integrated Approach to Recommend Citations in the Wild"
_TMLR — Under review for TMLR_

### Review · Reviewer_h7EA · 2026-06-16

**Summary Of Contributions:**

This paper studies local citation recommendation. The authors argue that existing methods mainly use local and global textual information, but do not fully use how a paper is perceived and cited by the research community. The paper proposes two main components. First, it introduces Profiler, a non-learnable and lightweight retrieval module. Profiler builds a “public profile” for each paper by combining the paper’s own representation with information from papers that cite it and the citation contexts where it is cited. Second, the paper proposes DAVINCI, a reranking model that combines semantic information with confidence scores from Profiler through an adaptive vector-gating mechanism. The proposed method outperforms several baselines such as BM25, SciNCL, HAtten, and SymTax. The experiments also show that Profiler is much faster than previous retrieval methods.

The main strengths of the paper are:
1. the problem is important and timely
2. the inductive evaluation setting is meaningful
3. the proposed Profiler is simple and scalable, and the experimental results are strong.

**Additional Comments:**

Overall, this paper is good writing and its research problem is important. Also, the proposed approach is appealing because it is simple, efficient, and easy to scale.

**Audience:**

Yes

**Audience Explanation:**

The paper is relevant to researchers working on information retrieval, recommender systems, scientific literature mining, and NLP for scholarly documents. Citation recommendation is an important practical task, and the paper studies it under a more realistic setting.

The introduced Profiler module may also be interesting to the TMLR audience because it is simple, scalable, and does not require training. This makes it potentially useful for large-scale scholarly search systems.

**Broader Impact Concerns:**

The paper argues that Profiler avoids some bias, but I do not think this concern is fully addressed. Since the method uses citation networks and citation contexts, it may still reflect existing patterns in our research community. I suggest that the authors add a broader impact discussion about this issue and clarify the possible risks of deploying citation recommendation systems in real scholarly writing tools.

**Claims And Evidence:**

Yes

**Claims Explanation:**

Overall, the claims are mostly supported by the experimental evidence. The paper provides results on multiple datasets and compares the proposed method with several strong baselines. The retrieval results show that Profiler improves both effectiveness and efficiency. The end-to-end results also show consistent gains over previous citation recommendation systems.

The ablation studies are useful. They show that removing the public profile hurts retrieval performance, and that different parts of DAVINCI contribute to the final performance. The analysis of different candidate pool sizes is also helpful.

**Requested Changes:**

1. Clarify and support the bias-related claims. This is my most important concern. The paper says that Profiler captures human citation patterns “without bias” or avoids confirmation bias. This statement is too strong, from my perspective. Since Profiler still uses citation links and citation contexts, it may still inherit existing citation bias, such as popularity bias, field bias, or bias toward well-known papers. The authors should either add experiments to measure bias or soften the claim.

2. Add more details about the inductive data construction. The inductive setting is a strong contribution, but the authors should explain more clearly how publication dates are obtained and handled. For example, what happens when the publication date is missing or ambiguous? Are preprints and conference versions handled consistently? This is important because temporal separation is central to the paper.

3.Make the comparison with large rerankers clearer. The paper compares DAVINCI with Qwen3-Reranker-8B and bge-reranker-v2-minicpm-40. This is a useful comparison, but the setup should be explained in more detail. The authors should describe the exact prompt, input format, truncation strategy, and whether any task-specific tuning was used. Without these details, it is hard to judge how fair the comparison is. Also, does the author consider the RankGPT[1]? Is it possible to compare with RankGPT for further analysis?

[1] Is ChatGPT Good at Search? Investigating Large Language Models as Re-Ranking Agents

---

> ### Author Response · Authors · 2026-07-11
> **Response to Reviewer 1 (h7EA)**
>
> We thank the reviewer for the constructive and encouraging feedback and for identifying our work as important and timely. Requested changes RC1 (bias-related claims) and RC2 (inductive data construction) are addressed in our common responses to both reviewers posted separately. We address the remaining requested changes below.
>
> **RC3 --- Comparison with large rerankers: details and RankGPT**
>
> *Implementation details for Qwen3-Reranker-8B and bge-reranker-v2-minicpm-40.* Kindly refer to Section 6.3 that already provides the exact prompt templates for both models and states that models are run in inference mode on same evaluation sets. We additionally clarify that truncation is applied to the document side when input exceeds the model's maximum context length, preserving the full query and citation context in all cases. We sincerely hope the clarification helps.
>
> *RankGPT comparison.* We have implemented and evaluated RankGPT's sliding window permutation generation strategy on three datasets using three open-source LLMs to have a detailed comparison: Mistral-7B-Instruct-v0.3, Llama-3-8B-Instruct, and Qwen3-8B. Closed-source APIs (e.g., GPT-4) are financially infeasible at our benchmark scale. All models use window size = 20, step = 10, Profiler's top-100 candidates as input as recommended in the RankGPT paper, and are evaluated on identical test sets and metrics as DAVINCI. Best values are shown in bold.
>
> **Comparing with RankGPT (instantiated with opensource LLMs):**
>
> | Model | Dataset | MRR | R@5 | R@10 | R@20 | NDCG@5 | NDCG@10 | NDCG@20 |
> |---|---|---|---|---|---|---|---|---|
> | Mistral-7B | ACL-200 | 32.49 | 42.19 | 51.34 | 67.68 | 32.85 | 35.76 | 39.90 |
> | Llama-3-8B | ACL-200 | 30.01 | 39.44 | 48.98 | 66.64 | 30.13 | 33.20 | 37.70 |
> | Qwen3-8B | ACL-200 | 36.50 | 46.62 | 54.98 | 70.16 | 37.17 | 39.85 | 43.66 |
> | **DAVINCI** | **ACL-200** | **48.92** | **62.08** | **70.36** | **76.46** | **50.92** | **53.62** | **55.17** |
> | Mistral-7B | FTPR | 37.53 | 48.72 | 58.03 | 74.74 | 38.19 | 41.18 | 45.40 |
> | Llama-3-8B | FTPR | 35.05 | 46.16 | 56.20 | 72.79 | 35.61 | 38.82 | 43.03 |
> | Qwen3-8B | FTPR | 44.96 | 57.47 | 65.16 | 79.03 | 46.38 | 48.86 | 52.34 |
> | **DAVINCI** | **FTPR** | **57.69** | **72.01** | **79.25** | **83.87** | **60.19** | **62.54** | **63.72** |
> | Mistral-7B | arXiv | 20.88 | 28.04 | 35.47 | 47.93 | 20.91 | 23.29 | 26.42 |
> | Llama-3-8B | arXiv | 18.93 | 26.18 | 33.93 | 46.99 | 18.88 | 21.36 | 24.67 |
> | Qwen3-8B | arXiv | 23.89 | 32.10 | 39.76 | 51.22 | 24.27 | 26.72 | 29.60 |
> | **DAVINCI** | **arXiv** | **28.85** | **39.02** | **47.65** | **55.33** | **29.89** | **32.69** | **34.64** |
>
> **Output stability (% of sliding windows with malformed permutations):** The repetition rate is the percentage of windows in which the model outputs duplicate identifiers (e.g., $[3] > [1] > [3]$), invalidating the permutation. The missing identifier rate is the percentage of windows in which some candidate identifiers never appear in the output at all, leaving those candidates unranked. A window can suffer from both simultaneously. Qwen3-8B's strong instruction-following keeps both rates near 1--2% except for arXiv, whereas Mistral-7B's missing identifier rate reaches 68.9% on arXiv --- nearly 7 in 10 windows produce incomplete rankings.
>
> | Model | Dataset | Repetition Rate (%) | Missing Identifier Rate (%) |
> |---|---|---|---|
> | Mistral-7B | ACL-200 | 36.7 | 32.5 |
> | Mistral-7B | FTPR | 35.4 | 34.7 |
> | Mistral-7B | arXiv | 37.0 | 68.9 |
> | Llama-3-8B | ACL-200 | 36.3 | 36.2 |
> | Llama-3-8B | FTPR | 39.8 | 36.6 |
> | Llama-3-8B | arXiv | 36.7 | 40.3 |
> | Qwen3-8B | ACL-200 | 1.6 | 1.5 |
> | Qwen3-8B | FTPR | 1.7 | 1.5 |
> | Qwen3-8B | arXiv | 1.7 | 5.6 |
>
> Two findings emerge, similar to RankGPT paper: (i) Mistral-7B and Llama-3-8B produce malformed permutations in 36--40% of windows --- more than a third of all reranking steps are unreliable; (ii) even Qwen3-8B, near-stable at ~1.7% repetition rate, is substantially outperformed by DAVINCI across all three datasets --- the MRR gap on ACL-200 alone is 12.42 points. DAVINCI's advantage holds even against the best available open-source reranking framework. We will add a dedicated subsection in Section 6 and narrow our architectural conclusion to: *"A supervised lightweight model with task-specific inductive biases outperforms zero-shot large-model baselines of up to $70\times$ more parameters in the citation recommendation setting"* --- consistent with the narrowing also requested by Reviewer 2 (RC4).
>
> **RC4 --- Broader impact on citation bias in deployment**
>
> We will add a broader impact paragraph to the conclusion acknowledging that citation-network-based methods may reflect existing community biases, and that deployment in real scholarly writing tools should be accompanied by periodic audits and, where feasible, diversity-aware reranking mechanisms. The quantitative bias analysis in our common response will be referenced here to ground this empirically.

---

### Review · Reviewer_tSNB · 2026-07-02

**Summary Of Contributions:**

This paper studies local citation recommendation (LCR) and makes a useful contribution by re-examining evaluation protocols for dynamically evolving scholarly corpora. In particular, the authors identify temporal-leakage risks in traditional transductive benchmarks and propose an inductive evaluation protocol with temporal precedence and disjoint data splits.

Methodologically, the paper proposes a decoupled architecture: Profiler pre-computes document "public profiles" from citation-network signals, and DAVINCI reranks retrieved candidates by combining Profiler-derived priors with semantic interaction. This is an interesting and practically motivated design for large-scale citation recommendation.

The key strengths are the paper's focus on a realistic temporal evaluation protocol, the scalable two-stage design, and the practical motivation for using citation-network context. The key weaknesses are that several central claims require stronger support: the temporal validity of Profiler's public profiles is unclear, the "bias-free" and "monotonic refinement" claims appear too strong, the single-run results lack uncertainty estimates, the efficiency comparison omits important end-to-end costs, and the large-model comparison does not isolate architecture, scale, and task-specific supervision.

**Audience:**

Yes

**Audience Explanation:**

he findings of this paper would likely interest at least some members of the TMLR audience. Local citation recommendation is practically important for managing the rapid growth of scientific literature and supporting scholarly writing. It is also relevant to researchers working on information retrieval, scientific AI, recommender systems, and natural language processing.

The paper's discussion of temporal leakage in traditional transductive evaluation is particularly valuable. The proposed inductive evaluation protocol is directly relevant to citation recommendation and may also inform other retrieval and recommendation tasks involving dynamically evolving corpora. This emphasis on benchmark rigor aligns well with the interests of the TMLR community.

The proposed architecture is also of interest because it explores how to combine graph-structural information with semantic interaction while maintaining computational efficiency on large academic graphs. Therefore, although the current version requires major revisions in its temporal masking clarification, theoretical claims, bias analysis, statistical evaluation, comparison design, and efficiency accounting, the problem setting, evaluation perspective, and preliminary framework are likely to stimulate useful discussion among relevant researchers.

**Broader Impact Concerns:**

I do not identify any major ethical concern that would require a dedicated Broader Impact Statement. The work focuses on academic citation recommendation. The main relevant concern is the standard recommender-system risk that citation-network-based methods may reinforce existing visibility or popularity biases, but this is already closely related to the technical concerns discussed above rather than a separate high-risk ethical issue.

**Claims And Evidence:**

No

**Claims Explanation:**

Although the paper reports numerical improvements across multiple datasets, the current evidence is not sufficient to support several of its strongest claims. The most serious issue is an ambiguity in the construction of Profiler under the proposed inductive setting. The paper defines an inductive protocol in which recommendations for a query paper should use only documents published before that query. However, Profiler constructs a candidate paper's public profile from its inward ego network, namely papers that cite the candidate paper, and the method is described as a one-off offline preprocessing step over the entire corpus. If these profiles include citing papers that are published after a given query, then the model would be using future citation information when scoring candidates for that query. This would undermine the central claim that the evaluation is temporally rigorous. If the authors do apply temporal masking during profile construction, this should be clearly stated and operationally described, since it also affects the claimed efficiency advantage.

The paper's claims about bias are also not adequately supported. The paper repeatedly states that Profiler avoids confirmation bias or mitigates popularity bias, but it does not provide direct measurements of bias in the recommendations. Because Profiler relies on inbound citation networks and citation contexts, it may inherit popularity, field, or historical biases present in those networks. Moreover, the base encoder is trained using citation-related signals, and highly cited papers may obtain richer public profiles simply because more citing papers and citation contexts are available. Degree-normalized averaging may reduce some forms of hub amplification, but it does not by itself establish that the resulting recommendations are unbiased. The paper should provide stratified analyses on long-tail papers, papers with different citation counts, newly published papers, and interdisciplinary papers, or otherwise soften the claim substantially.

The "monotonic refinement" and "never degrades" claims are similarly stronger than the current evidence supports. Mean aggregation over citing papers may improve representations in many cases, but noisy citing papers or topic drift could plausibly degrade a candidate's base semantic representation. The paper currently provides intuition and aggregate empirical improvements, but not a theoretical guarantee or targeted empirical boundary conditions showing when the profile enrichment is safe. This claim should either be supported with a clearer proof and assumptions, or rephrased as an empirical observation.

The statistical reliability of the main experiments and the fairness of the comparison setup also limit the persuasiveness of the conclusions. The paper states that the final results are based on a single comprehensive run, but it does not report error bars, confidence intervals, or statistical significance tests. As a result, the current evidence cannot fully rule out random variation when claiming comprehensive improvements over prior methods.

More importantly, the paper uses its results to argue that task-specific architectural design is more important than raw parameter scale. However, the relevant comparisons contrast a supervised, task-trained lightweight model with larger general-purpose rerankers used mainly in zero-shot or prompted settings. This conflates task-specific supervision, zero-shot generalization, architecture, and model scale. The current experimental setup therefore does not cleanly support the broader conclusion about architecture versus scale.

Finally, the efficiency claim needs a more complete accounting. The reported retrieval time appears to focus on online retrieval, while Profiler also requires offline construction of profiled representations and indexing over the corpus. If temporal masking is required for the inductive protocol, this preprocessing may no longer be a simple one-off operation. The paper should report end-to-end costs, including profile construction, indexing, retrieval, reranking, memory usage, and throughput.

Overall, the paper presents a potentially useful engineering approach, but it frames the contribution as a strong methodological and benchmarking advance without yet providing sufficiently rigorous, reproducible, leakage-free, and fair evidence to support its strongest claims.

**Requested Changes:**

### Critical to Acceptance

1. Please clarify whether each candidate paper's public profile is constructed with temporal masking relative to each query. Profiler uses the inward ego network of a candidate paper, i.e., papers that cite it, and the method section describes profiling as a one-off offline preprocessing step over the entire corpus. If citing papers published after the query are included in a candidate's profile, the system would use future citation information and violate the proposed inductive protocol. If temporal masking is applied, please explain exactly how this is implemented and whether the reported efficiency numbers include the corresponding cost.

2. The paper states several times that Profiler avoids confirmation bias or is "bias-free." I do not think the current experiments support this claim. Because Profiler relies on inbound citation networks, citation contexts, and an encoder trained with citation-related signals, it may still capture popularity, field, or historical biases. Highly cited papers may also receive richer profiles simply because more citation contexts are available. I recommend adding direct bias analyses, such as comparing the citation-count distribution of recommended papers against the ground-truth distribution, and stratifying results by citation count, paper age, long-tail status, field, or interdisciplinary status. If such analyses are infeasible, the claim should be softened. A more defensible statement would be that Profiler does not explicitly use citation counts or venue prestige as input features, rather than that it is fully unbiased.

3. The paper claims that Profiler "never degrades" below semantic retrieval and continues to improve as structural evidence grows. This currently reads more like a design intuition than a strict theoretical guarantee. Noisy citing papers or topic drift could plausibly degrade the base semantic representation. Unless the authors can provide a theoretical proof with clear assumptions, I recommend rephrasing this claim more cautiously, for example: "We empirically observe that Profiler generally improves performance in our experiments."

4. The paper argues that task-specific architectural design can outperform raw parameter scale. However, the current comparison places a supervised, task-trained DAVINCI model against larger general-purpose rerankers used in zero-shot or off-the-shelf settings. This comparison mixes the effects of task-specific supervision and zero-shot generalization, so it does not cleanly establish a conclusion about parameter scale versus architecture. To support the stronger claim, the larger models would need to be fine-tuned on the same training data under comparable conditions. Otherwise, the conclusion should be restricted to a narrower statement, such as: "A supervised lightweight model outperforms zero-shot or off-the-shelf large-model baselines in this setting."

5. The paper should report the full end-to-end cost of the system. Retrieval time is reported in detail, but the complete system also includes offline profile construction, indexing, reranking, memory usage, and throughput. This is especially important if temporal masking is required for the inductive setting, because profile construction may no longer be a simple one-off preprocessing step.

6. The ablations for DAVINCI are currently reported only on smaller datasets such as ACL-200 and FTPR. Since a major part of the paper's argument concerns scalability and performance on large datasets such as arXiv and ArSyTa, this leaves an important gap. Please add ablations on at least one large-scale dataset, or include a clear explanation of why such experiments are not feasible.

### Would Strengthen the Work

1. It would be helpful to add statistical significance tests or uncertainty estimates. Since the main results come from a single full run, the authors could report confidence intervals or repeat experiments on representative subsets.

2. More qualitative analysis would be useful. The current case study is helpful, but a single example is not enough to understand the model's behavior across different situations. Additional success and failure cases, especially examples where Profiler introduces misleading neighborhood signals, would make the analysis more informative.

3. The paper should more clearly separate claims about performance under the proposed inductive benchmark from broader claims of general superiority over prior systems. Some prior baselines were originally designed for transductive settings, so if the authors want to make stronger claims about overall method superiority, an additional sanity check under the original transductive setting or a more careful discussion of this comparison would be helpful.

4. The paper should clarify the implementation details needed for reproducibility, including how citation context snippets are extracted, how the inward ego network is bounded, how missing full-text documents are handled, and how profile construction is performed under temporal constraints.

---

> ### Author Response · Authors · 2026-07-11
> **Response to Reviewer 2 (tSNB)**
>
> We thank the reviewer for the rigorous review and pointing us towards the right directions too. The requested changes are binned to Critical to Acceptance (CA) and Would Strengthen (WS) groups. CA1 (temporal validity), CA2 (bias claims), and WS4 (reproducibility details) are addressed in our common responses to both reviewers posted separately. We address the remaining concerns below.
>
> **CA3 --- "Never degrades" claim**
>
> The reviewer is correct that this reads as a stronger guarantee than we can formally prove. It was a design intuition and empirical results verified it on an aggregate scale but we do not have a formal proof covering all boundary conditions or a per-sample guarantee. Following the reviewer's recommended rephrasing, we will cautiously replace "never degrades" and "monotonic refinement" language with:  "We empirically observe that Profiler generally improves performance in our experiments."
>
> **CA4 --- Comparing large models**
>
> We agree. Our comparison pits a lightweight task-trained reranker against large general-purpose rerankers used in a  zero-shot, instruction-prompted setting. As the reviewer prescribes, we restrict our conclusion to "A supervised lightweight model outperforms zero-shot or off-the-shelf large-model baselines in this setting." The RankGPT experiments reported in Reviewer 1's response (RC3) further support this narrowed conclusion.
>
> **CA5 --- End-to-end efficiency accounting**
>
> The "Comp. Time" in Table 2 covers both phases of Profiler's pipeline together: the offline phase of building entire profiled corpus from training set, and the online phase of constructing valid $D_{\text{eval}}$ by retaining only queries whose ground-truth cited paper exists in $C$, followed by multi-GPU cosine similarity search over the profiled vectors. No per-query re-computation is needed. Reranking times are not reported for any system including DAVINCI because SOTA baselines (HAtten, SymTax) share architectural commonalities with DAVINCI at the reranking stage and we do not claim computational superiority there. Reporting retrieval times uniformly across all systems ensures a fair comparison of the component where our approach differs most substantially. We will add a clarifying note to Section 4.2 making this explicit.
>
> **CA6 --- Ablations on large-scale datasets**
>
> Full DAVINCI ablations on Refseer, arXiv or ArSyTa require training a separate model for each ablation variant on millions of citation contexts which is computationally prohibitive given linear scaling with both context count and candidate pool size. We offer two partial mitigations. First, the retrieval-stage ablation in Table 4 already covers all five datasets: on arXiv, Recall@10 is 33.4 with profiling versus a maximum attainable 28.9 without --- a consistent gap that demonstrates Profiler's scalability. Second, Table 5's DAVINCI ablations are designed to isolate architectural contributions rather than demonstrate scalability, and two datasets of different sizes and domains (ACL-200 and FTPR) are not exhaustive but sufficient for this purpose. We will clearly separate the purpose of Table 4 (scalability evidence for Profiler) from Table 5 (architectural analysis of DAVINCI) in Section 5.
>
> **WS1 --- Statistical significance**
>
> Single-run evaluation is standard practice in the LCR literature at this scale --- SymTax and HAtten, the closest baselines, also report single-run results, as multiple training runs on datasets of this size are computationally prohibitive. Our results are therefore directly comparable with prior work under the same convention. As noted in Section 4.1, we conducted a stability analysis on representative subsets prior to the full run and observed minimal variance, which provided sufficient confidence to proceed with a single comprehensive run.
>
> **WS2 --- Additional qualitative examples**
>
> We will try adding two further case studies w.r.t retrieval: one success case where the public profile signal rather than direct semantic matching drives the correct retrieval, and one failure case where Profiler introduces a misleading neighbourhood signal, e.g., a candidate paper cited predominantly in a different subfield whose profile drifts toward that subfield's vocabulary, causing it to be incorrectly ranked for a query from the candidate's original field. This directly relates to the reviewer's concern about topic drift in citing contexts.
>
> **WS3 --- Comparison Framing**
>
> Prior systems did not consciously design for transductive evaluation --- strict temporal separation was a blind spot in the evaluation community that our paper identifies and corrects. We will revise the framing in Section 4.3: *"Prior systems were not designed with transduction as a deliberate choice; temporal separation was an overlooked constraint in existing benchmarks. Our inductive protocol corrects this oversight. Comparisons in Table 3 are best interpreted as performance under a more demanding and realistic evaluation setting."*

---

### Review · Reviewer_2GUK · 2026-07-18

**Summary Of Contributions:**

This submission addresses the task of local citation recommendation by proposing two main components, alongside a modified evaluation protocol:

Profiler: An offline method that enriches candidate-paper representations by incorporating the text of their citing papers and corresponding citation contexts.

DAVINCI: An online reranker that combines semantic text matching with an initial retrieval prior via a vector gating mechanism.

Evaluation Protocol: A temporally constrained, inductive evaluation setup designed to prevent data leakage across time.

Key Strengths
Leveraging citation contexts provides an intuitive and localized signal for recommendation while keeping online retrieval efficient via vector similarity search.

The empirical evaluation is extensive, spanning five datasets of varying scales and demonstrating solid improvements in metrics like MRR, Recall, and NDCG.

The paper provides deep-dive analyses into several components, such as score transformation, gating, and pool sizes.

Key Weaknesses
The execution of the temporal protocol is unclear, leaving open questions about potential data leakage in candidate profiles.

Several strong claims—specifically regarding the system being "bias-free" and the flaws of transductive evaluation—lack direct empirical support.

Baselines and ablations for the DAVINCI reranker do not properly control for model capacity or task-specific fine-tuning.

**Audience:**

Yes

**Audience Explanation:**

Recommendation systems, document retrieval, and "AI for Science" are highly active areas of interest for the TMLR community. Specifically, researchers working on automated literature reviews, digital libraries, and graph-augmented language models would find value in the insights regarding how local citation contexts can be vectorized offline to improve dense retrieval without increasing online computational overhead. Furthermore, the discussion surrounding temporal evaluation protocols in scientific corpora is highly relevant to establishing robust benchmarking practices in information retrieval.

**Broader Impact Concerns:**

There are no major ethical or dual-use concerns that strictly necessitate a standalone Broader Impact Statement. However, because the paper explicitly touches upon mitigating "bias" in scientific discovery, the authors should be mindful that citation recommendation engines fundamentally shape scientific visibility. If a recommendation system implicitly favors specific citation networks or fails to account for structural biases in who gets cited, it can inadvertently gatekeep scientific exposure. Tempering the "bias-free" claim (as requested above) sufficiently addresses this algorithmic fairness consideration.

**Claims And Evidence:**

Yes

**Claims Explanation:**

While the broad empirical gains across multiple datasets suggest that the proposed pipeline is effective, several core claims lack rigorous support:

Temporal Leakage: The paper claims to follow a strict inductive temporal protocol. However, it is unclear if a candidate paper’s profile changes depending on the query date. If a candidate uses citations that appeared after a specific test query's publication date, the temporal protocol is violated.

"Bias-Free" Claim: The assertion that Profiler eliminates confirmation or popularity bias is unsupported. Simply averaging neighbor representations does not guarantee the graph is free from underlying prestige or visibility biases, and the paper lacks an operational definition or metric to prove this claim.

Critique of Transductive Evaluation: The authors argue that transductive evaluation inflates performance, but they only point to differences in dataset scale rather than running a controlled experiment comparing the same model under both protocols.

Confounding Factors in Reranking: The superiority of DAVINCI's gating architecture over massive models like Qwen3-Reranker is attributed to design, but DAVINCI was explicitly fine-tuned for this task while the larger baselines were only prompt-adapted. Similarly, the ablations do not control for parameter capacity.

Data Quality: Clear data contradictions in the tables (e.g., conflicting NDCG values for FullTextPeerRead and ArSyTa) undermine the clarity and reliability of the reported evidence.

**Requested Changes:**

Critical Changes (Required for Acceptance)

Clarify the Temporal Cutoff Mechanism: Explicitly state whether candidate profiles are dynamically restricted based on the query paper's publication date. Verify and mathematically/procedurally demonstrate that no post-query citations are leaked into the candidate representations during test time.

Address Numerical Inconsistencies: Correct the conflicting data points across tables. Specifically, fix the FullTextPeerRead NDCG@10 discrepancy between Table 3 ($64.68$) and Table 5 ($64.48$), and the ArSyTa at $k=300$ NDCG@5 discrepancy between Table 6 ($34.73$) and Table 3 ($24.73$). Review all other tables for alignment.

Tone Down or Substantiate the "Bias-Free" Claim: Either provide a direct evaluation metric tracking citation/popularity bias or rephrase the claim to reflect reality—i.e., that Profiler avoids explicitly weighting by raw citation counts, rather than claiming it is inherently "bias-free."

Report Statistical Significance: Provide confidence intervals, standard deviations, or paired bootstrap significance tests for the main results in Table 3. This is essential given the narrow margins of victory over baselines like SymTax on datasets like RefSeer and arXiv.

Strengthening Changes (Recommended but Optional)

Isolate Architecture vs. Capacity for DAVINCI: Run a controlled baseline comparison for the vector gating mechanism against standard feature concatenation or a size-matched MLP to prove the architectural choice itself drives performance, not just added parameters.

Control for Fine-Tuning in Reranker Baselines: Compare DAVINCI against a parameter-matched BGE or Qwen baseline that has been similarly fine-tuned on the task data, rather than relying solely on prompted zero-shot configurations.

Directly Validate the Transductive Critique: Run a mini-experiment evaluating the same model architecture under both a transductive protocol and your proposed inductive protocol to definitively prove that transductive setups overestimate performance.

Expand Ablations Across Datasets: Include the component ablations (currently limited to ACL-200 and FullTextPeerRead) for RefSeer, arXiv, and ArSyTa—especially since profiling showed minimal impact on ArSyTa.

Document Hyperparameters and Preprocessing fully: Provide explicit details on how the final $k=300$ was selected despite varying optima in Figure 1. Document the exact decay parameters, publication date sources/granularity, and how duplicate or missing records were handled.